# Successful Desensitization to Sorafenib and Imatinib—A Report of Two Cases and a Literature Review

**DOI:** 10.3390/healthcare12060601

**Published:** 2024-03-07

**Authors:** Natasa Kusic, Vesna Tomic Spiric, Snezana Arandjelovic, Aleksandra Peric Popadic, Ivana Bozic Antic, Milan Dimitrijevic, Rada Miskovic, Ljiljana Stefanovic, Aleksandra Plavsic

**Affiliations:** 1Clinic of Allergy and Immunology, University Clinical Centre of Serbia, 11000 Beograd, Serbia; vesna.tomic-spiric@med.bg.ac.rs (V.T.S.); snezana.arandjelovic2019@gmail.com (S.A.); popealeksandra@yahoo.com (A.P.P.); milandimitrijevic1411@gmail.com (M.D.); rada.miskovic@med.bg.ac.rs (R.M.); ljstefan.md@gmail.com (L.S.); sandrony@yahoo.com (A.P.); 2Faculty of Medicine, University of Belgrade, 11000 Belgrade, Serbia; 3Euromedik Healthcare System, 11158 Belgrade, Serbia; dr.ivana.bozic@gmail.com; 4Faculty of Dentistry Pancevo, University Business Academy in Novi Sad, 21000 Novi Sad, Serbia

**Keywords:** desensitization, drug hypersensitivity reaction, imatinib, sorafenib

## Abstract

Background: Drug desensitization allows for safe administration of a drug to a patient with a previous hypersensitivity reaction. Successful desensitization protocols have been described for different medications, including protocols for oncology patients. Few cases of desensitization to sorafenib and imatinib have been described in the literature so far. Objective: The objective of this paper is to describe the process of the sorafenib and imatinib drug hypersensitivity diagnosis and desensitization process in two patients. Methods: Two oncology patients who experienced non-immediate hypersensitivity reactions to sorafenib and imatinib underwent desensitization to these drugs. We designed a protocol for the first patient and used a modified protocol from the literature for the second patient. Results: By using a slow desensitization technique and gradual tapering of corticosteroids and antihistamines, both patients reached the target dose of the incriminated drug. Conclusions: Desensitization to sorafenib and imatinib can be an effective therapeutic option in patients with hypersensitivity to those medications, without alternative treatment options.

## 1. Introduction

Drug hypersensitivity reactions (DHRs) represent adverse effects of drugs that may vary in clinical presentation and outcome. With the increasing availability of a wide variety of medications, the frequency of DHRs is also increasing. According to Coombs and Gell’s classification, there are four different types of DHRs: type I (IgE-mediated), type II (antibody-mediated cytotoxicity reactions), type III (immune-complex-mediated) and type IV (delayed hypersensitivity). However, the growing understanding of DHRs has led to new classification that incorporates phenotypes, endotypes and biomarkers [1]. There are two phenotypes based on the clinical presentation and timing of DHRs: immediate drug onset allergy (occurring 1 to 6 h after drug exposure) and delayed drug onset allergy (occurring days after drug exposure and presenting itself as an isolated reaction, with one organ or multiorgan involvement). The defined endotypes include IgE-mediated, Aspirin exacerbated respiratory disease (AERD) and a Human leukocyte antigen (HLA) associated drug hypersensitivity reaction.

According to the European Academy of Allergy and Clinical Immunology (EAACI), the usual practice after the occurrence of a DHR is to permanently avoid the culprit drug and to use alternative, non-cross-reacting drugs whenever it is possible [2]. In situations when the incriminated drug is necessary for optimal future treatment, when patients have limited accessibility to an alternative drug or when an alternative drug is not available, physicians may consider drug desensitization, but the risk–benefit ratio needs to be assessed carefully. Drug desensitization is defined as the induction of a temporary state of tolerance of a compound responsible for a hypersensitivity reaction [3]. It is a high-risk procedure of administering increasing doses of the medication under continuous medical supervision until the total cumulative therapeutic dose is achieved and tolerated [3]. This procedure was developed in patients who presented IgE/non-IgE type I hypersensitivity reactions. For delayed DHRs, desensitization is restricted to mild, uncomplicated exanthema and fixed drug reactions [2]. Fixed drug eruption (FDE) is a type of drug-induced eruption, in which intraepidermal CD8+ T cells in the lesional skin are the final effector cells in an epidermal injury. The presence of CD25+CD4+ T cells in the epidermis of FDE lesions may be involved in the induction of desensitization to FDE. Desensitization can be effective in treating FDE [2]. Successful desensitization protocols have been described for different agents, most commonly for antibiotics, biologics, antineoplastic drugs, progesterone and aspirin/nonsteroidal anti-inflammatory drugs [4]. Few cases of desensitization to sorafenib and imatinib have been described in the literature so far [5,6,7,8,9,10,11].

Sorafenib is a multikinase inhibitor with antiproliferative and antiangiogenic effects. It is an anticancer drug approved for the treatment of unresectable hepatocellular carcinoma and advanced renal cell carcinoma, but it is also used for the treatment of advanced thyroid cancers [12,13]. The most common side effects to sorafenib occur particularly in the cardiovascular and gastrointestinal systems; however, different cutaneous reactions have been described as well [14]. The most frequently occurring cutaneous effects are hand–foot skin reactions (HFSRs), uncharacterized skin eruption, subungual splinter hemorrhage, alopecia, pruritus, dry skin and flushing [15]. Typical delayed-type cutaneous hypersensitivity reactions are not common; there have only been a few reports of sorafenib-induced erythema multiforme and drug reaction with eosinophilia and systemic symptoms (DRESS) syndrome [15,16,17,18]. The management of severe cutaneous adverse reactions (SCARs) requires discontinuation of the incriminated drug.

Imatinib is a tyrosine kinase inhibitor that is used for hematological malignancies, gastrointestinal stromal tumors (GISTs) and systemic mastocytosis. Cutaneous adverse reactions to imatinib are very common; their frequency ranges from 7 to 88.9% [6]. Most of the skin reactions are dose-dependent and are related to direct pharmacological effects of the drug. Some of them are alopecia, dyschromia, erythema, HFSRs and pruritus. Other cutaneous conditions are caused by immune-mediated hypersensitivity, IgE- or cell-mediated [7]. These conditions may occur any time since the introduction of imatinib generally consists of urticaria, angioedema, maculopapular rash, Stevens Johnson syndrome (SJS) and DRESS syndrome. Since there is no specific allergy testing proven to diagnose imatinib hypersensitivity, and there is a possibility of overlapping in the clinical presentation of dose-dependent and hypersensitivity reactions, a careful allergological evaluation is necessary in order to undertake further therapeutic steps [7].

We present two patients who developed hypersensitivity reactions to sorafenib and imatinib and were successfully treated with desensitization. 

## 2. Materials and Methods

### 2.1. Report of Cases 

#### 2.1.1. Patient 1

A 38-year-old man was diagnosed with non-iodine-avid disseminated papillary thyroid cancer. After unsuccessful treatment using adriamycin chemotherapy, programmed death 1 inhibitor pembrolizumab and gemcitabine chemotherapy, fourth-line therapy with sorafenib (2 × 400 mg/daily) was initiated. Ten days after the beginning of the therapy the patient experienced skin changes and a low-grade fever. A dermatological examination revealed facial erythema, erythematous macules and oral mucosal lesions. Laboratory evaluation showed normal complete blood count but pathological hepatogram: aspartate aminotransferase (AST) 54 (0–37 U/L); alanine aminotransferase (ALT) 62 (0–41 U/L). Sorafenib therapy was discontinued, and the patient was given oral corticosteroid and antihistamine therapy, as well as topical emollients. Skin changes completely resolved within a few weeks. Then, sorafenib therapy with reduced dose (200 mg/daily) was restarted. One hour later, the patient developed generalized erythema and facial swelling with a fever of 40.0 °C. According to the Registry of Severe Cutaneous Adverse Reactions (RegiSCAR), this reaction was classified as a possible DRESS case (score 2) [19]. The patient was given parenteral corticosteroids and antihistamines. Sorafenib therapy was stopped and the symptoms improved. Since there were no other treatment options, an allergist referred the patient for desensitization.

#### 2.1.2. Patient 2

A 64-year-old woman was treated with imatinib, 400 mg/daily, for GIST. Within eleven days from imatinib introduction, she developed eyelid swelling and generalized erythema. The dose of imatinib was reduced to 300 mg/daily; however, skin changes were still progressing. The examination revealed generalized maculopapular exanthema, oral enanthema and eyelid edema. Imatinib therapy was discontinued. The patient was treated with systemic and topical corticosteroids and antihistamines until the complete resolution of skin changes. Since imatinib was the only therapy for GIST treatment, she was referred to our Clinic for desensitization after four months. 

Before the beginning of the desensitization procedure, the risk–benefit ratio for both patients was assessed carefully. Since there were no alternative treatment options and patients did not experience severe, life-threatening reactions to sorafenib and imatinib, we decided to proceed with desensitization. The patients were required to sign informed consent. Previously, they were adequately informed about potential side effects and hypersensitivity reactions during the procedure. A complete physical examination and laboratory analysis were performed before the desensitization procedure. The protocols were conducted in a hospital setting, under continuous monitoring by experienced personnel. Skin and mucous membranes, peak expiratory flow and blood pressure were monitored during the procedure. The patients were cannulated with a peripheral intravenous line. Emergency equipment and necessary medications for potential anaphylaxis treatment were available at any time. Since there is no universal drug desensitization protocol, we designed a protocol for the first patient, while a modified protocol by Penza et al. [6] was used for the second patient. The protocols for both patients are shown in Table 1 and Table 2.

## 3. Results

### Desensitization Procedure and Outcome 

In the first case, we started the protocol with 50 mg of sorafenib with premedication using corticosteroids and antihistamines (Table 1). Over eleven days, we gradually increased the dose of sorafenib to 200 mg. At the same time, the dose of corticosteroid was slowly reduced. During the desensitization protocol, there were no breakthrough reactions. At the time of hospital discharge, the cumulative dose of 200 mg was reached, and we advised the patient to increase the dose by50 mg every two weeks to achieve the maximum tolerable dose of 2 × 350 mg. Corticosteroids and antihistamines were slowly discontinued. For the next six months, the patient was treated with the target dose of 2 × 350 mg of sorafenib without adverse reactions.

Regarding patient 2, after a successful desensitization protocol, a tolerance of 400 mg of imatinib was achieved. A 100 mg capsule of imatinib was dissolved in glycerin until concentrations of 1 and 10 mg/mL were reached. Serial dilutions were then formulated, using sterile water as diluent. We started with the lowest dose, 10 ng/mL concentration, based on the protocol by Penza et al. [6], with a slow dose increase. We believed that this was the safest way to reach a good outcome, even with a breakthrough reaction. On the sixth day, at the dose of 100 mg, the patient experienced pruritus, palm erythema and eyelid swelling. Corticosteroid therapy and antihistamines were given with gradual tapering. After 21 days, the cumulative dose of 400 mg was reached, and the patient continued using imatinib without adverse reactions. 

## 4. Discussion

Protein kinase inhibitors are increasingly utilized in the treatment of various types of malignancies. With the increased usage of these agents, a diverse array of cutaneous toxicities associated with these drugs has been described. While most skin reactions are non-life threatening, severe cutaneous adverse reactions may also occur. They include SJS, toxic epidermal necrolysis (TEN), DRESS and acute generalized exanthematous pustulosis (AGEP) [20]. These drugs are toxic by nature. Most skin reactions are usually related to the pharmacological effects of the drug and are taken as a positive indicator of drug effectiveness [8]. However, DHRs can also occur.

DHRs to imatinib and sorafenib have been poorly investigated, and very few cases of desensitization to these drugs have been reported. Nelson et al. reported 10 patients with leukemia and rash associated with imatinib that underwent desensitization [9]. After the complete desensitization procedure, four patients had no skin changes, four developed a rash that was treated successfully with corticosteroids and antihistamines and two were not able to resume therapy due to are current rash that occurred 5 h and several days after the desensitization protocols. The authors concluded that in leukemic patients with imatinib-associated rash, desensitization may be helpful. Our patient developed a delayed DHR less than 2 weeks after imatinib was started. While the patient did not respond to the dose reduction, the drug discontinuation and applied therapy led to rash resolution. In the case report by Penza et al., a patient developed hypersensitivity reaction 8 weeks after imatinib therapy, which turned into diffuse pruritic erythema that persisted with corticosteroids and dose reduction, but resolved when therapy was stopped [6]. The patient underwent a 24-day desensitization protocol, and his initial positive skin prick test became negative after desensitization. Regarding sorafenib, we found only two reports of sorafenib desensitization in the literature [5,8]. Linauskiene et al. described the case of a 21-year-old woman who developed fever and urticaria 10 days after the start of sorafenib therapy for metastatic lamellar hepatocellular carcinoma in a dosage of 800 mg daily [5]. The desensitization was performed using premedication (20 mg of prednisolone and antihistamines), reaching the cumulative dose of 2 × 400 mg between the fifth and seventh days. The patient developed urticaria on the eighth day; the dose was reduced to 400 mg in the morning and 200 mg in the evening, and sorafenib therapy was tolerated without skin changes. In the second case report of a 68-year-old woman, a DHR occurred 2 weeks after the introduction of 800 mg of sorafenib. Since the therapy was urgently needed for metastatic renal cancer, the desensitization was performed in one day reaching the dose of 798 mg in 3 h, using premedication with corticosteroid an hour before the start of the process [8]. The patient experienced generalized erythema one hour after the test was performed. In the next 6 days, sorafenib was given in increased doses (2 × 100 mg, 2 × 200 mg, 4 × 200 mg with 2 h intervals in between each consecutive dose) and the final dose of 2 × 400 mg was reached. The corticosteroid was slowly stopped and antihistamines were used as needed. The patient successfully continued sorafenib therapy. In our two cases, patients were referred to our Clinic when their skin reactions had already been resolved.

According to medical documentation and anamnestic data, we assessed these reactions as delayed DHRs to imatinib and sorafenib. Since there were no other therapeutic options, we decided to proceed with desensitization according to EAACI recommendations. There is no universal drug desensitization protocol for delayed-type hypersensitivity reactions. Protocols may vary in duration, starting dose and the time interval between doses. It is highly desirable to reach the therapeutic dose as quickly as possible in the safest way. Also, there is no consensus on the dose of premedication before and during desensitization. The dose and the type of premedication are usually based on the patient’s symptoms during the initial hypersensitivity reaction [21]. Antihistamines and corticosteroids have often been used in premedication, and they might be associated with higher rates of successful desensitization and shorter desensitization period. However, premedication may not always prevent breakthrough reactions, which in some cases are more severe than the initial reaction [2]. Before the beginning of premedication with corticosteroids in our two cases, the risk–benefit ratio for both patients was assessed. Bearing in mind the initial presentation of hypersensitivity reactions in our patients, as well as the fact that imatinib and sorafenib were the only treatment options for them, we decided to use higher doses of corticosteroids, regardless of their side effects. We believed that the possible side effects of corticosteroid use for both patients were less significant than desensitization failure. However, in both patients, no side effects of corticosteroid use were recorded. According to the EAACI position paper, slower protocols tend to be more effective for delayed reactions [2]. Case reports of successful desensitization in patients with severe cutaneous reactions have been presented in the literature, including sorafenib [5,8] and imatinib, with rapid [9] and slow protocols [6,7,10,11]. We chose a slow protocol for both patients, considering the delayed occurrence of their clinical symptoms. Because of the potential severe initial hypersensitivity reaction in the case of patient number 1, we decided to premedicate him before every dose during desensitization. During the procedure, there were no breakthrough reactions, and we were able to reach the target dose and slowly discontinue corticosteroids and antihistamines. In case number 2, we did not use premedication at the beginning of the protocol. However, on the sixth day, when the patient experienced palmar erythema, pruritus and eyelid swelling, we treated her with prednisolone and antihistamines. The corticosteroids and antihistamines led to symptom resolution, and we decided to continue the procedure. The dose of 400 mg was reached, and the patient continued this therapy without corticosteroids and antihistamines.

The exact mechanisms of desensitization processes are still not completely understood. Several mechanisms have been proposed: high-affinity IgE receptor (FcεRI) internalization, anti-drug IgG4 blocking antibody, altered signaling pathways in mast cells and basophils, and reduced Ca^2+^influx, all of them leading to inhibition of the activation of mast cells and basophils [4,22,23,24]. The mechanisms regarding delayed DHR desensitization should be further explored [2]. In the study by Teraki et al., it was demonstrated that the number of lesional CD4+ CD25+ T cells in FDE is increased significantly after desensitization, whereas the number of lesional CD8+ T cells decreased. It is suggested that the CD25+ CD4+ T cells found in the epidermis of FDE lesions after desensitization might have a regulatory function, thereby suppressing the effector function of CD8+ T cells in FDE lesions. The CD25+CD4+ T cells in the epidermis of FDE lesions after desensitization are likely the result of the continual migration of CD25+CD4+ T cells into the epidermis due to repeated administration of the drug [25]. In the study by Klaewsongkram et al., the authors investigated the proportion of drug-induced CD4+CD25+CD134+ T-cell changes in the peripheral blood of patients with a history of imatinib-induced non-immediate reactions undergoing drug desensitization. CD134, or OX40, a member of the tumor necrosis factor receptor family, is characterized as a costimulatory molecule regulating both TH1- and TH2-mediated reactions and has a critical role in the maintenance of an immune response. The co-expression of CD134 and CD25 after 48 h of antigen stimulation can be used as a marker for antigen-specific CD4+ T cells. The authors reported successful desensitization protocols, and observed a reduction in imatinib-induced CD4+CD25+CD134+ T-cell percentages in peripheral blood after tolerance induction in these patients. It remained low even after discontinued use of steroids [10]. The authors suggested that the process of drug desensitization in non-immediate hypersensitivity may lead to a diminished drug-specific T-cell response. In everyday practice, it is very important to assess the DHR, to consult an allergist and, when it is necessary, to make a decision about desensitization, taking into account the risks and benefits.

## 5. Conclusions

Early diagnosis and proper assessment of skin reactions are crucial after DHRs. An individual approach should be applied to each patient to reach the most optimal solution, including desensitization. This is especially important for oncology patients who have limited accessibility to alternative drugs. Desensitization to sorafenib and imatinib should be considered in patients with DHRs who do not have any other treatment options with carefully planned protocols.

## Figures and Tables

**Table 1 healthcare-12-00601-t001:** Desensitization protocol to sorafenib.

Day	Dose, mg	Premedication, mg	Reaction
1	50	methylprednisolone 40 iv. + desloratadine 5 p.o.	Ø
2	50	methyprednisolone 30 iv. + desloratadine 5 p.o.	Ø
3	75	methyprednisolone 30 iv. + desloratadine 5 p.o.	Ø
4	100	methyprednisolone 30 iv. + desloratadine 5 p.o.	Ø
5	100	methyprednisolone 30 iv. + desloratadine 5 p.o.	Ø
6	150	methyprednisolone 30 iv. + desloratadine 5 p.o.	Ø
7	200	methyprednisolone 30 iv. + desloratadine 5 p.o.	Ø
8	200	methyprednisolone 20 iv. + desloratadine 5 p.o.	Ø
9	200	methyprednisolone 15 iv. + desloratadine5 p.o.	Ø
10	200	methyprednisolone 10 iv. + desloratadine 5 p.o.	Ø
11	200	prednisone tbls 10 p.o. + desloratadine5 p.o.	Ø

Abbreviations: iv.—intravenous; p.o.—per os; tbls—tablets.

**Table 2 healthcare-12-00601-t002:** Desensitization protocol to imatinib.

Day	Concentration/ Dosage	Volume (mL)/Dosage(Caps)	Cumulative Dose, mg	Premedication, mg	Reaction	Therapy, mg
1	10 ng/mL 100 ng/mL 1 µg/mL 10 µg/mL 100 µg/mL	1, 2, 4 mL 1, 2, 4 mL 1, 2, 4 mL 1, 2, 4 mL 1, 2, 4 mL	0.000077 0.00077 0.0077 0.077 0.77	Ø	Ø	Ø
2	1 mg/mL	1, 2, 4 mL	7	Ø	Ø	Ø
3, 4, 5	10 mg/mL	1, 2, 4 mL	70	Ø	Ø	Ø
6, 7, 8	100 mg	1 cap	100	Ø	palmar erythema, pruritus, eyelid edema	methylprednisolone 40 iv. + chloropyramine 20 im. + ranitidine 50 iv.
9, 10, 11	200 mg	2 caps	200	prednisone 20 p.o. + bilastine 2 × 20 p.o.	Ø	Ø
12, 13, 14	300 mg	3 caps	300	prednisone 10 p.o. + bilastine 2 × 20 p.o	Ø	Ø
15, 16, 17	350 mg	3 +1/2 caps	350	prednisone 10 p.o. + bilastine 2 × 20 p.o	Ø	Ø
18, 19, 20, 21	400 mg	4 caps	400	prednisone 10 p.o. + bilastine 2 × 20 p.o	Ø	Ø

Abbreviations: caps—capsules; im.—intramuscular; iv.—intravenous; p.o.—per os.

## Data Availability

Data are available on request from the authors.

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
