# Peer review of "Successful Desensitization to Sorafenib and Imatinib—A Report of Two Cases and a Literature Review"

_healthcare, 2024, doi:10.3390/healthcare12060601_

Round 1

Reviewer 1 Report

Comments and Suggestions for Authors

The described cases are interesting both for oncologists and allergists. As desensitization is used too seldom, in my opinion these cases are worthy being published. The proposed novel desensitization protocol for sorafenib may be useful for others. 

Could the authors explain how they obtained the lowest doses of imatinib used on the first five days? Did they use a parenteral form or dilute oral form in a solvent? It needs to be explained if this protocol is to be useful for other allergists. 

My another concern is related to relatively high doses of methylprednisone used i-v on the daily basis in the first case. What was the rationale for such high doses of this drug? The authors discussed briefly this problem, but this problem needs more attention and more detailed explanation.      

Comments on the Quality of English Language

There are some small language mistakes such as e.g.: "an allergist was consulted" (p. 2, line 96) or "the patients were provided an intravenous route" (p. 3, line 117).

Reviewer 2 Report

Comments and Suggestions for Authors

Major:

Lines 48-49: drug desensitization in fixed drug eruptions? In which situations?

Lines 64-65: are the authors referring to SCARs? Do the authors manage SCARs with dose reduction?

The Case reports belong in the methods section.

Line 94-95: the authors should quantify the RegiSCAR for this reaction and classify it accordingly.

Cases must be  equally described (imatinib is rather less detailed)

The authors ought to discuss the reasons why they decided to start both desensitization protocols with premedication with glucocorticoids and the implications this act has on the desensitization mechanism.

Minor

Lines 41-43: which "certain situations"? Be more precise.

Lines 210-214: the mentioned mechanisms are all regarding immediate DHR. The authors should replaced these with delayed DHR desensitization mechanisms.

Comments on the Quality of English Language

The English language requires a thorough revision. Some phrases are unintelligible, with several plural words missing the "s", and almost all phrases lack definite articles (i.e., "the").

Reviewer 3 Report

Comments and Suggestions for Authors

The author must provide follow-up data from an ethics committee and informed consent for use of patient-associated data. Since the use of an ethics committee is necessary when a research study involves the participation of human subjects, either directly or through the use of their biological material or medical records.

It should be discussed more extensively regarding the use of steroids and the effects that they themselves can produce in the withdrawal of this as it is associated significantly to secondary skin reactions such as acne or rashes and other more significant disorders that may be associated with the drugs used.

It should make clear why the side effects of steroid use are not considered and what is expected in people with hypersensitivity reactions under this context and sorafenid and imatinib.

Round 2

Reviewer 2 Report

Comments and Suggestions for Authors

The authors have correctly addressed most of my comments.

There is one minor detail that still requires clarification, as it seems contradictory:

- in line 53 the authors mention the importance of CD4+ CD25+ T cells in dessensitization, but bellow on line 239, it is mentioned that CD4+CD25+CD134+ T cell counts tend to decrease thanks to desensitization.

Please clarify, taking into account the true function of CD4+CD25+ T cells, specifically CD134+ cells.

Author Response

Thank you very much for taking the time to review this manuscript, for recognizing the content of our article, and for allowing us to make changes according to your suggestions. Please find the detailed responses below and the corrections that were made, highlighted in green in the re-submitted files.

  1. In line 53 the authors mention the importance of CD4+ CD25+ T cells in desensitization, but bellow on line 239, it is mentioned that CD4+CD25+CD134+ T cell counts tend to decrease thanks to desensitization. Please clarify, taking into account the true function of CD4+CD25+ T cells, specifically CD134+ cells.
    Response: Thank you for the comment. In a study by Teraki et al, it was demonstrated that the number of lesional CD4+ CD25+ T in fix drug eruptions (FDE) is increased significantly after desensitization, whereas the number of lesional CD8+ T cells decreased. It is suggested that the CD25+ CD4+ T cells found in the epidermis of FDE lesions after desensitization, might have a regulatory function, thereby suppressing the effector function of CD8+ T cells in FDE lesions. The CD25+CD4+ T cells in the epidermis of FDE lesions after desensitization are likely the result of continual migration of CD25+CD4+ T cells into the epidermis due to repeated administrations of the drug (reference 25).
    On the other hand, in the study by Klaewsongkram et al, the authors investigated the proportion of drug –induced CD4+CD25+CD134+ T-cell changes in peripheral blood in patients with a history of imatinib-induced non-immediate reactions undergoing drug desensitization. The CD134, or OX40, a member of the tumor necrosis factor receptor family, is characterized as a costimulatory molecule regulating both TH1- and TH2-mediated reactions and has a critical role in the maintenance of an immune response. The coexpression of CD134 and CD25 after 48 hours of antigen stimulation can be used as a marker for antigen-specific CD4+ T-cells. The authors reported a successful desensitization protocols, and observed a reduction in imatinib-induced CD4+CD25+CD134+ T-cell percentages in peripheral blood after tolerance induction in these patients. It remained low even after discontinued use of steroids. (reference 10).
    We have added this in lines 238-255.
